# Lead-Free Copper-Based Perovskite Nanonets for Deep Ultraviolet Photodetectors with High Stability and Better Performance

**DOI:** 10.3390/nano12193264

**Published:** 2022-09-20

**Authors:** Shuhong Xu, Jieqin Tang, Junfeng Qu, Pengfei Xia, Kai Zhu, Haibao Shao, Chunlei Wang

**Affiliations:** 1Advanced Photonics Center, School of Electronic Science and Engineering, Southeast University, Nanjing 210096, China; 2School of Electronics & Information, Nantong University, Nantong 226019, China

**Keywords:** Cs_3_Cu_2_I_5_, photoelectric detection, ultraviolet, nanonet

## Abstract

Considering practical application and commercialization, the research of non-toxic and stable halide perovskite and its application in the field of photoelectric detection have received great attention. However, there are relatively few studies on deep ultraviolet photodetectors, and the perovskite films prepared by traditional spin-coating method have disadvantages such as uneven grain size and irregular agglomeration, which limit their device performance. Herein, uniform and ordered Cs_3_Cu_2_I_5_ nanonet arrays are fabricated based on monolayer colloidal crystal (MCC) templates prepared with 1 μm polystyrene (PS) spheres, which enhance light-harvesting ability. Furthermore, the performance of the lateral photodetector (PD) is significantly enhanced when using Cs_3_Cu_2_I_5_ nanonet compared to the pure Cs_3_Cu_2_I_5_ film. Under deep ultraviolet light, the Cs_3_Cu_2_I_5_ nanonet PD exhibits a high light responsivity of 1.66 AW^−1^ and a high detection up to 2.48 × 10^12^ Jones. Meanwhile, the unencapsulated PD has almost no response to light above 330 nm and shows remarkable stability. The above results prove that Cs_3_Cu_2_I_5_ nanonet can be a great potential light-absorbing layer for solar-blind deep ultraviolet PD, which can be used as light absorption layer of UV solar cell.

## 1. Introduction

As an important photoelectric device that can convert incident light into electrical signals, photodetectors (PD) have a wide range of applications in target detection, astronomy, imaging, environmental sensing and optical communications [1,2,3,4,5]. Moreover, compared with visible light and near-infrared light detectors, ultraviolet (UV) photodetectors have vital applications in military fields such as missile defense and tracking, and secure space-to-space communication [6,7,8,9,10,11]. So far, many ultraviolet photodetectors have been prepared using some wide band gap semiconductor materials, such as graphene, Al_x_Ga_1__–__x_N, Mg_x_Zn_1–x_O, SnO_2_, TiO_2_, Zn_2_GeO_4_, GaN, etc. [10,12,13,14]. However, due to the relatively wide direct band gap, low quantum efficiency, slow response speed, complex manufacturing process and high cost, the practical application of these materials is limited in optoelectronic devices [3,15].

Fortunately, in recent years, hybrid halide perovskites have been widely considered due to their excellent photophysical properties, including adjustable band gap, intense light absorption, charge-carrier diffusion lengths and high charge-carrier mobilities [16,17,18,19,20]. Moreover, great progress has been made in exploring the application of perovskite as a light absorber in photovoltaic devices, photodetectors and light-emitting diodes [21,22,23]. For ultraviolet light photodetectors, Maculan et al. used the reverse temperature crystallization method to grow bulk CH_3_NH_3_PbCl_3_ single crystals, and designed the first UV photodetector based on this single crystal, which has a high on-off current ratio, fast light response and long-term light stability [24]. Cho and coworkers have greatly improved the emission of blue CsPbCl_3_ nanocrystals (NCs) through the plasmon effect of Ag film and photonic crystal effect of polymethyl methacrylate (PMMA) opal photonic crystals (OPCs). They constructed a high-performance flexible UV photodetector employing CsPbCl_3_/Ag/OPCs hybrids, which has low dark current of 10^−11^ A, high detection sensitivity of 9 × 10^14^ Jones, short response time of 28 ms/31 ms and narrow response line width of 30 nm [25]. Unfortunately, the environmental instability and lead toxicity of photodetectors based on halide perovskites have hindered their practical applications and future commercialization. In response to the goals of carbon peaking and carbon neutralization, some progress has also been made in the exploration of lead-free perovskite systems recently, such as CsSnX_3_ [26], Cs_2_SnI_6_ [27], Cs_3_Bi_2_Br_9_ [28], Cs_3_Sb_2_Br_9_ [29], Cs_2_AgBiX_6_ [30], CsCu_2_X_3_, [31] and Cs_3_Cu_2_X_5_ [32]. After comparison, it is found that Cs_3_Cu_2_I_5_ NCs have a high photoluminescence quantum yield (PLQY), strong deep ultraviolet absorption and excellent stability, making it an ideal candidate for UV photodetection materials.

At the same time, structural factors such as crystal structure, grain size, pattern arrangement and hierarchical structure have a great influence on its optoelectronic performance [33]. At present, some researches on the performance of Cs_3_Cu_2_I_5_ perovskite UV photodetectors are mainly concentrated on its planar films. For instances, Shi et al. prepared stable deep-ultraviolet photodetectors based on solution-processed Cs_3_Cu_2_I_5_ film and Cs_3_Cu_2_I_5_/GaN heterojunction [34,35]. Moreover, many studies on the construction of patterned micro- and nano-structure arrays have been proven to manage light effectively [36]. In addition, there have been related reports to improve the performance of photodetectors by constructing 2D ordered perovskite nanostructure arrays with a relatively simple method. Qi et al. prepared high-quality CH_3_NH_3_PbI_3_ nanonet and nanobowl arrays on different substrates by nanosphere lithography and fabricated a photodetector successfully using perovskite nanonet [37]. Zhou and coworkers obtained the ordered nanostructured CsPbBr_3_ nanofilm grown in the preferred (110) direction through a similar method, and prepared a photodetector on this basis [38]. However, so far, there has been no report on the preparation of ordered Cs_3_Cu_2_I_5_ perovskite nanostructure arrays using this simple nanosphere-lithography technique. The effect of Cs_3_Cu_2_I_5_ arrays on the performance of Cs_3_Cu_2_I_5_ UV photodetectors has not been studied.

Here, we self-assembled a large-area 2D MCC on the air-water surface by a simple method. Then, a high-quality and non-toxic Cs_3_Cu_2_I_5_ film with an ordered nanonet structure was prepared, based on the MCC template. Due to the unique structure, the light capture ability of the film was significantly enhanced, which was beneficial to the performance of the equipment. Cs_3_Cu_2_I_5_ nanonet lateral PD constructed by 1 μm PS spheres exhibits a high photoresponsivity of 1.66 A W^−^^1^ and a high detectivity of 2.48 × 10^12^ Jones, which is much better than the traditional spin-coated film PD. In addition, the device shows excellent stability against air, heat and UV without encapsulation. Moreover, Cs_3_Cu_2_I_5_ UV photodetectors are very sensitive to ultraviolet light and hardly respond to visible light. Therefore, the Cs_3_Cu_2_I_5_ nanonet as a light-absorbing layer can become a great potential candidate for solar-blind ultraviolet PD.

## 2. Experimental Section

Materials: Monodisperse PS spheres (1 μm, 5 wt% in water) were purchased from Suzhou Sunway Biotechnology Co., Ltd. (Suzhou, China). Dimethyl sulfoxide (DMSO, 99.5%), copper Iodide (CuI, 99.9%, AR) and Sodium dodecyl sulfate ((CH_3_(CH_2_)_11_OSO_3_, SDS) were purchased from Aladdin (Shanghai, China). Cesium Iodide (CsI, 99.9%, AR) was purchased from Maclin. The glass plate with ITO interdigital electrodes was purchased from Advanced Election Technology Co., Ltd. (Yingkou, China); the finger width was 50 μm, the pitch was 50 μm and the electrode thickness was 150 nm. All materials were used without any further treatment.

**Preparation of the monolayer colloidal crystal (MCC) template:** First, the monodisperse PS sphere solution, ethanol and deionized water were mixed in a volume of 1:1:1, centrifuged at 3000 rpm for 10 min and the supernatant discarded, repeated three times, and dispersed the precipitate in a mixture of deionized water and ethanol. Then, the mixture was dispersed on the surface of the deionized water in the glass petri dish until it covered the entire water surface to form a relatively dense color MCC film. After a few hours, small amounts of SDS were dropped onto the water surface to pack the MCC film closely. Subsequently, the MCC film was picked up with an ultraviolet ozone treated 1.5 × 1.5 cm^2^ ITO/glass plate with ITO interdigital electrodes, through inserting the plate into the solution at an oblique angle. Finally, the plate coated by the MCC film was annealed at 90 °C for 5 min.

**Preparation of the Cs_3_Cu_2_I_5_ nanonet photodetectors:** The 0.75 mol/L perovskite precursor was prepared by dissolving CsI and CuI with a molar ration of 3:2 completely at 75 °C in DMSO and then was filtered through a 536 PTFE 0.22 μm filter. A volume of 80 μL precursor was spin-coated on the prepared 1.5 × 1.5 cm^2^ MCC template at 500 rpm for 10 s and then 2500 rpm for 40 s, and baked on a hot plate at 100 °C for 15 min. After that, it was immersed into toluene for 5 min to remove the PS spheres. Finally, the substrate was annealed at 100 °C for 1 h to form a nanonet with high crystallinity and uniformity.

**Preparation of the traditionally spin-coated Cs_3_Cu_2_I_5_ film photodetectors:** The 0.75 mol/L perovskite precursor was prepared by dissolving CsI and CuI with a molar ration of 3:2 completely at 75 °C in DMSO and then filtering through a 536 PTFE 0.22 μm filter. After that, 80 μL precursor was spin-coated on an ultraviolet ozone treated 1.5 × 1.5 cm^2^ glass plate with ITO interdigital electrodes at 500 rpm for 10 s and then 2500 rpm for 40 s, and baked on a hot plate at 100 °C for 1 h.

**Characterization:** UV-visible (UV-vis) absorption spectra were recorded with a Shimadzu 3600 UV-vis near-infrared spectrophotometer (Shimadzu, Japan). Photoluminescence (PL) spectra were recorded with a Shimadzu RF-5301 PC spectrofluorometer excited at 280 nm. The morphology of the devices was characterized by scanning electron microscope (SEM) and atomic force microscopy (AFM). The X-ray powder diffraction (XRD) was carried out to characterize the phase. The electrochemical impedance spectroscopy (EIS) was measured by an electrochemical workstation. The photodetection was tested via home-built system integrated into a monochromator and Keithley 2440 Source Meter. As-prepared PD was directly tested in the open air at room temperature. For current-voltage (I-V) measurement under photo or under dark, PD with area of 1.5 × 1.5 cm^2^ was exposed under the 275 nm light irradiation or in a dark room.

## 3. Results and Discussion

The main manufacturing process of the Cs_3_Cu_2_I_5_ nanonet photodetector is schematically shown in Figure 1. Firstly, in a glass petri dish filled with ionized water, a water/ethanol dispersion containing monodisperse PS colloidal spheres with a size of 1 μm was tightly spread over the water surface to form a MCC film. The film was then transferred the MCC film to the ITO/glass substrate, and annealed at 90 °C for five minutes. The Cs_3_Cu_2_I_5_ precursor solution was spin-coated on the template prepared above, and annealed at 100 °C for 10 min. Finally, after immersing in toluene for 15 min, PS colloidal spheres were removed. After annealing at 100 °C for 1 h, the Cs_3_Cu_2_I_5_ nanonet was formed on the ITO/glass substrate.

The top-view SEM image and its enlarged illustration in Figure 2a show the close-packed MCC film of 1 μm PS microspheres self-assembled on the surface of ITO, which was used as the template. After spin-coating the Cs_3_Cu_2_I_5_ precursor solution and removing the PS microspheres, the Cs_3_Cu_2_I_5_ nanonet was formed with nanoholes as shown in Figure 2b. As revealed by the SEM images and the inset, the diameters of the Cs_3_Cu_2_I_5_ nanoholes were about 757.3 nm, slightly shorter than the diameter of PS spheres (1165 nm). From AFM in Figure 2c,d, the size of the pores from AFM in Figure 2c was very consistent with the SEM observation in Figure 2b. Moreover, the thickness of the nanonet was about 600 nm, which is smaller than the diameter of the PS microspheres. It suggests the observed Cs_3_Cu_2_I_5_ are actually spherical segments with height higher than that of the halfsphere. As a result, it is comprehensible for the observed diameter of the pore smaller than 1 μm since the maximal pore diameter appears at the half heigh of PS spheres. The AFM result in Figure 2d provides a direct proof for the bowl morphology inside the nanonet since the side wall of the bowl is inclined to the substrate. Appendix A shows the energy dispersive X-ray spectroscopy (EDS) at the nanonet and at the nanoholes. As expected, there were also signals of Cs, Cu and I in the EDS for the nanohole. This also confirms the observed nanonet is a bowl-like net.

The crystal structure of Cs_3_Cu_2_I_5_ nanonet was confirmed by XRD. To make a comparison, the compact films of Cs_3_Cu_2_I_5_ were prepared by spin-coating of the precursors on glass without using MCC template (SEM image of Cs_3_Cu_2_I_5_ compact films in Appendix A). As shown in Figure 3a, the diffraction patterns for Cs_3_Cu_2_I_5_ nanonet and compact film are well coincident with the standard diffractions of orthorhombic Cs_3_Cu_2_I_5_ (PDF#45-0077), suggesting the formation of Cs_3_Cu_2_I_5_ crystals in the nanonet and compact film. Figure 3b shows the steady-state photoluminescence spectra of nanonet and compact film. Under excitation by a 280 nm laser, Cs_3_Cu_2_I_5_ nanonet and compact films exhibit a broad as well as symmetric emission peak at 442 nm with Full Width at Half Maximum (FWHM) of 75 nm. Furthermore, the UV–vis absorption spectra of the Cs_3_Cu_2_I_5_ nanonet and compact films in Figure 3c show the absorption edge around 330 nm. It can be seen that the hollow nanonet-array structure represents much stronger absorbance than that of the compact spin-coated film. This result is probably attributed to the enhanced light scattering inside the pore of the nanonet structure and larger area, giving nanonet a much higher absorption efficiency than the compact film. Moreover, a broad peak around 600 nm can be observed for the nanonet. It may be due to the increased thickness [37].

In order to study the potential application of this Cs_3_Cu_2_I_5_ nanonet structure in the field of ultraviolet photoelectric, a lateral structure Cs_3_Cu_2_I_5_ deep ultraviolet photodetector was constructed, with the geometry as seen in Figure 4a. To fabricate such a photodetector, the MCC floating on the water was transferred to a glass substrate which was predefined with ITO interdigital electrodes with a width and pitch of 50 μm; the thickness of the ITO electrode was 150 nm. Then the Cs_3_Cu_2_I_5_ crystalline was directly grown on MCC which was removed by toluene solution. To quantify the performance of the photodetector, I-V characteristics were measured under an incident light of 275 nm with light intensity ranging from 0 to 2.17 mW/cm^2^. Figure 4b reveals that photocurrent gradually increases with the increase of light intensity and the linear I-V behavior indicates typical ohmic contact between ITO and perovskite. Compared with the traditional planar perovskite photodetector, the photocurrent of the nanonet structure is significantly higher at the same light intensity of 0.095 mW/cm^2^, as seen in Figure 4c. In addition, the relationship of photocurrent and light intensity can be described by the power law: $I_{P}=\alpha P^{\beta }$, where $I_{P}$ is photocurrent, P represents the light intensity and α and β are the proportional constant and the R-squared coefficient, respectively. Figure 4d shows the fitting curve under 5 V bias and β is calculated to be 0.98803, which reflects the conversion efficiency of incident light energy is relatively stable. Furthermore, we measured EIS of the photodetectors with different film structures. As seen in Figure 4e, the diameter of the semicircle corresponding to nanonet is smaller than compact film from the Nyquist plot at the high-frequency region. This indicates that the charge transport resistance of photodetector with the MCC template is relatively small, which can improve carrier transport dynamics. In addition, we calculated other important parameters characterizing the photodetector, including the responsivity (R), specific detectivity (D*) and external quantum efficiency (EQE), to further evaluate its performance. R, D* and EQE are given by the following equations [39]:(1)$$R=\frac{I_{P}-I_{d}}{P_{opt}S}$$
(2)$$D^{*}=R\sqrt{\frac{S}{2eI_{d}}}$$
(3)$$EQE=\frac{I_{p}/e}{P_{opt}/hv}=R\frac{hc}{e\lambda }$$
where $I_{P}$, $I_{d}$, $P_{opt}$ and S represent the photocurrent, the dark current, the incident light intensity and the effective illuminated area, while e, h, c and *λ* are the elementary charge, the Planck’s constant and the speed of light and the light wavelength (275 nm), respectively.

In Figure 4f,g, at the bias voltage of 5 V, the responsivity, specific detectivity and EQE show a non-linear decline trend with the increase of light intensity. The biggest responsivity, detectivity and EQE of the nanonet photodetector are 1.66 A W^−1^, 2.48 × 10^12^ Jones (1 Jones = 1 cm Hz^1/2^ W^−1^) and 750%, respectively, under the incident light intensity of 3.8 × 10^−^^3^ mW/cm^2^, which are larger than the compact film structure (Appendix A). The improved light harvest due to scattering of incident light inside the nanonet is believed as the reason for the better performance of nanonet PD. Similar results were also reported for CH_3_NH_3_PbI_3_ nanonet PD and film PD[37]. In Appendix A, the tested response time for nanonet PD and film PD is similar, with the rise time around 9 s and decay time around 6 s. Besides, Figure 4h,i display the I–t curves of the nanonet photodetector operated at different bias voltages and light intensities. It can be seen clearly that with the increase of the light intensity or bias voltages, the photocurrent increases distinguishably, which reveals that under different illumination and bias conditions, the photocurrent of the PD is stable during repeated on/off cycles. Besides, the stability of Cs_3_Cu_2_I_5_ nanonet PD was estimated by putting PD under either thermal environment at 120 °C or strong ultraviolet light (275 nm, 3.5 mW/cm^2^) environment for 0 h, 6 h and 12 h. After that, the nanonet photodetector was tested. From I-t curves of Appendix A, the current of PD can preserve 97.8% (6 h) and 94.9% (12 h) in the 120 °C thermal environment or 97% (6 h) and 88% (12 h) in the ultraviolet light environment in comparison to that of freshly-fabricated PD. Obviously, Cs_3_Cu_2_I_5_ nanonet photodetector has high stability.

## 4. Conclusions

In summary, a Cs_3_Cu_2_I_5_ nanonet lateral photodetector that is sensitive to deep ultraviolet light and shows excellent stability has been successfully constructed based on a monolayer colloidal crystal template. Compared with the conventional Cs_3_Cu_2_I_5_ planar thin-film photodetector, the performance of this device is significantly better, showing a high light responsivity of 1.66 A · W^−1^ and a high detection of 2.48 × 10^12^ Jones. The innovation of nanonet structure for Cs_3_Cu_2_I_5_ photodetectors provides a promising way to obtain higher performance non-toxic deep ultraviolet photoelectric devices. Moreover, this nanonet structure of Cs_3_Cu_2_I_5_ can be used in an ultraviolet solar cell field.

## Figures and Tables

**Figure 1 nanomaterials-12-03264-f001:**
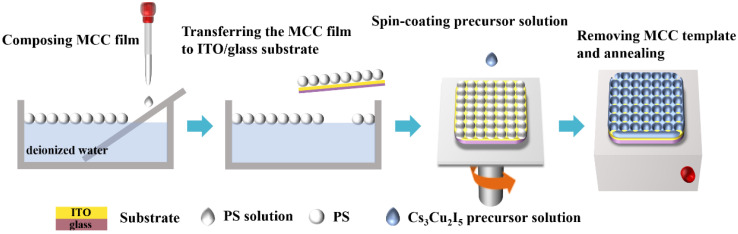
Processing procedure for the preparation of the Cs_3_Cu_2_I_5_ nanonet photodetector.

**Figure 2 nanomaterials-12-03264-f002:**
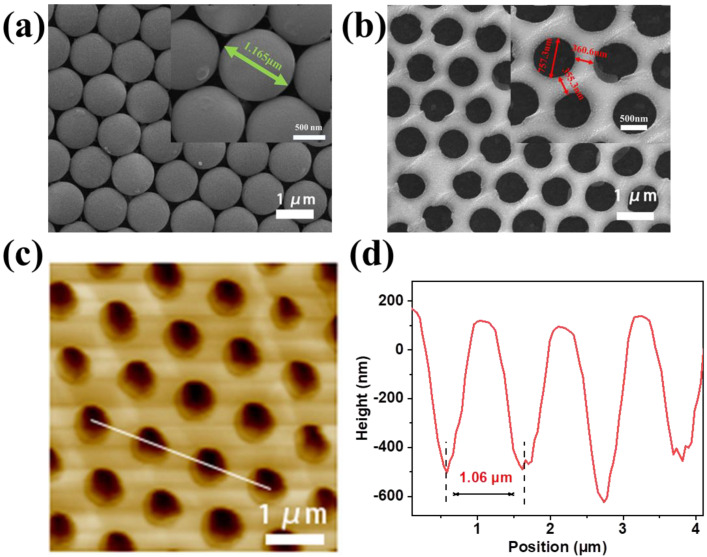
(**a**,**b**) The top-view SEM images of PS spheres and Cs_3_Cu_2_I_5_ nanonet. The inset shows the corresponding enlarged SEM images. (**c**,**d**) AFM image and line profiles of Cs_3_Cu_2_I_5_ nanonet.

**Figure 3 nanomaterials-12-03264-f003:**
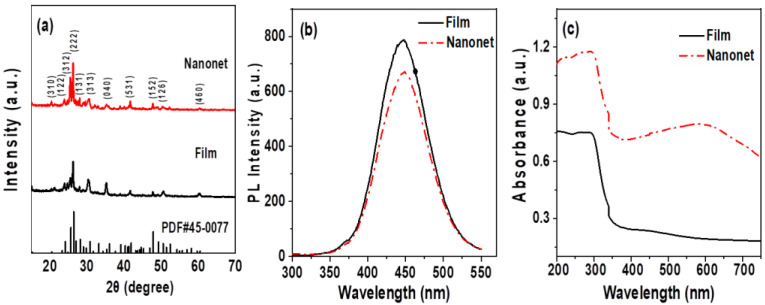
(**a**) XRD patterns of the Cs_3_Cu_2_I_5_ standard data, Cs_3_Cu_2_I_5_ nanonet and compact films. (**b**) PL spectra and (**c**) UV–vis absorption spectra of the Cs_3_Cu_2_I_5_ nanonet and compact film.

**Figure 4 nanomaterials-12-03264-f004:**
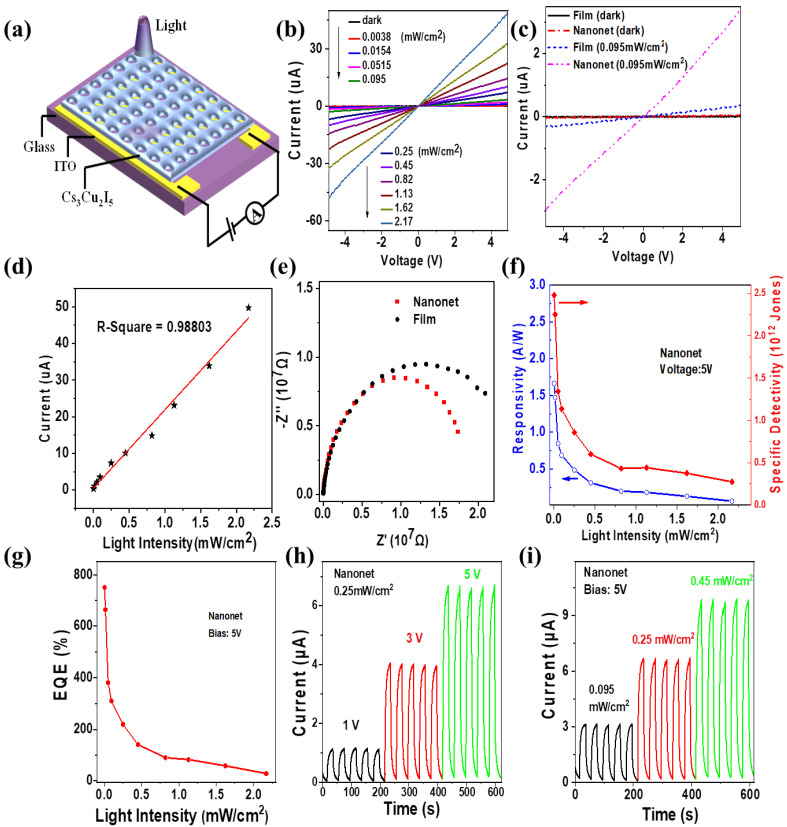
(**a**) Schematic illustration of the photodetector based on Cs_3_Cu_2_I_5_ nanonet. (**b**) I–V curves of the Cs_3_Cu_2_I_5_ nanonet photodetector under 275 nm irradiation with varied intensities and in the dark. (**c**) I–V curves of the Cs_3_Cu_2_I_5_ nanonet and film photodetector under 275 nm irradiation with 0.095 mW/cm^2^ and in the dark. (**d**) Logarithmic plot of the photocurrent versus light irradiation intensities at a bias of 5 V. Each star is Cs_3_Cu_2_I_5_ nanonet photodetector under different light intensity. (**e**) The typical Nyquist plots of nanonet and film photodetector in high frequency region. (**f**) Responsivity and detectivity of the nanonet photodetector versus the light density. The biggest responsivity is 1.66 A · W^−^^1^, the highest detectivity is 2.48 × 10^12^ Jones. (**g**) The EQE of the nanonet photodetector versus the light density. The highest EQE is 750%. I–t curves of the nanonet photodetector operated at different (**h**) bias voltages and (**i**) light intensities.

## Data Availability

Not applicable.

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
