# Peer review of "Lead-Free Copper-Based Perovskite Nanonets for Deep Ultraviolet Photodetectors with High Stability and Better Performance"

_nanomaterials, 2022, doi:10.3390/nano12193264_

Round 1
Reviewer 1 Report
General
The plots are too small and the captions are barely visible. I would suggest to increase the size.
Supplementary material: I would suggest to put the supplementary material into the main paper. In my view, it doesn’t make much sense to move one photo and two plots into another file. Btw, it is something wrong with axis titles fonts in the Figure S2
Detailed
Title: … and performance of deep ultraviolet.. (of is missing)
L22/23: please, unbreakable space between the number and units, e.g. 330 nm. I would also suggest to use SI units throughout the paper.
L 197-199: Please, give numbers to the equations.
L 202: wavelength(275nm) -> please, use spaces, i.e. wavelength (275 nm)
L227-228: Please, don’t use symbol ~. It looks strange given a two digits precision
References: I didn’t understand the use of [J] after each paper title.
L275, Ref [17] at the very end: 59-+ . What means -=+?
Author Response
Dear Editor Ada Liu and Reviewer: Thanks for your E-mail about the manuscript of “High stability and performance deep ultraviolet photodetectors based on lead-free copper-based perovskite nanonets” (nanomaterials-1914385). We have made revisions according to the reviewer’s and the editor’s comments. We hope the current version of manuscript satisfying the requirement of Nanomaterials. All revisions are marked in red in the revised manuscript.
Sincerely yours,
Prof. Chunlei Wang
Advanced Photonics Center, School of Electronic Science and Engineering, Southeast University, Nanjing, 210096 (P. R. China)
E-mail: wangchl@seu.edu.cn
Thank you for all constructive suggestions. We have revised the manuscript according to these helpful suggestions.
Comment 1: The plots are too small and the captions are barely visible. I would suggest to increase the size.
Reply: We have changed the size of all figures.
Comment 2: Supplementary material: I would suggest to put the supplementary material into the main paper. In my view, it doesn’t make much sense to move one photo and two plots into another file. But, it is something wrong with axis titles fonts in the Figure S2
Reply: Since we complemented more figures in the supplementary material, so we keep original Figure S1 and S2 in the supplementary material. In addition, the fonts for all figures, including Figure S2 has been revised.
Comment 3: Title: … and performance of deep ultraviolet.. (of is missing)
Reply: Title of the manuscript has been changed into “Lead-free copper-based perovskite nanonets for deep ultraviolet photodetectors with high stability and better performance”
Comment 4: L22/23: please, unbreakable space between the number and units, e.g. 330 nm. I would also suggest to use SI units throughout the paper.
Reply: We have changed unbreakable space between the number and units. And We changed some units to SI units.
Comment 5: L 197-199: Please, give numbers to the equations.
Reply: we have added (1), (2) and (3) for these equations.
Comment 6: L 202: wavelength(275nm) -> please, use spaces, i.e. wavelength (275 nm)
Reply: We have changed unbreakable space between the number and units in the whole manuscript.
Comment 7: L227-228: Please, don’t use symbol ~. It looks strange given a two digits precision
Reply: The sentence of “showing a high light responsivity of ~1.66 AW-1 and a high detection of ~2.48 × 1012 Jones” has been changed into “showing a high light responsivity of 1.66 AW-1 and a high detection of 2.48 × 1012 Jones”.
Comment 8: References: I didn’t understand the use of [J] after each paper title.
L275, Ref [17] at the very end: 59-+ . What means -=+?
Reply: We have revised all the References according to requirement. The symbol [J] and -+ have been removed.

Reviewer 2 Report
Critical reading is required including figure legends formatting (Figure S2 and others), biggest responsivity, etc. The response time can also be shown in supporting or main section to demonstrate the speed of a photodetector.
Cs3Cu2I5 nanonet lateral photodetector performed better than conventional Cs3Cu2I5 planar thin-film photodetector, more stability and high responsivity and detectivity values. Why this is happened is not explained clearly to the general readership.
The experimental details are an important part of the manuscript so all the details should be added to the manuscript for the scientific community to follow the critical parameters. Some details are missing such as what is area of exposure, dark conditions, experimental conditions vacuum or air etc.
The reference template is not uniform such as some references show month and year of publication, but some don’t (ref1 and others if required). One template should be followed to remove these non-uniformities. The authors are also required to cite recent papers on photodetection including Advanced Functional Materials 32 (3), 2105038 (2022) and Nanotechnology 31 (15), 152002 (2020)
Author Response
Dear Editor Ada Liu and Reviewer: Thanks for your E-mail about the manuscript of “High stability and performance deep ultraviolet photodetectors based on lead-free copper-based perovskite nanonets” (nanomaterials-1914385). We have made revisions according to the reviewer’s and the editor’s comments. We hope the current version of manuscript satisfying the requirement of Nanomaterials. All revisions are marked in red in the revised manuscript.
Sincerely yours,
Prof. Chunlei Wang
Advanced Photonics Center, School of Electronic Science and Engineering, Southeast University, Nanjing, 210096 (P. R. China)
E-mail: wangchl@seu.edu.cn
Thank you for all constructive suggestions. We have revised the manuscript according to these helpful suggestions.
Comment 1: Critical reading is required including figure legends formatting (Figure S2 and others), biggest responsivity, etc. The response time can also be shown in supporting or main section to demonstrate the speed of a photodetector.
Reply: We have added the corresponding information in figure legends. The response time have been complemented in Figure S4 in the supporting information.
Revision: (1) The caption of figure S3 (namely original figure S2) has revised as following: “(a) Responsivity and detectivity of the compact film photodetector versus the light density. The biggest responsivity is 0.185A W−1, the highest detectivity is 0.48 × 1012 Jones. (b) The EQE of the compact film photodetector versus the light density. The highest EQE is 85%”
(2) Some caption of figure 4 has also been revised as following: “(f) Responsivity and detectivity of the nanonet photodetector versus the light density. The biggest responsivity is1.66 A W−1, the highest detectivity is 2.48 × 1012 Jones. (g) The EQE of the nanonet photodetector versus the light density. The highest EQE is 750%.” has been added in Figure 4f and g.
(3) The response time have been complemented into Figure S4 in the supporting information. In the second paragraph of page 8, the following discussion has been complemented: “In figure S4 in the supporting information, the tested response time for nanonet PD and film PD is similar with the rise time around 9 s and decay time around 6 s. ”
Comment 2: Cs3Cu2I5 nanonet lateral photodetector performed better than conventional Cs3Cu2I5 planar thin-film photodetector, more stability and high responsivity and detectivity values. Why this is happened is not explained clearly to the general readership.
Reply: Since Cs3Cu2I5 nanonet are formed in the interspace between monolayer colloidal crystal (MCC) templates fabricated by using 1μm polystyrene (PS) spheres, each void hole in the nanonet likes a bowl. AFM result in figure 2d provides a direct proof for the bowl morphology since the side wall of the bowl is inclined to the substrate. In the revised manuscript, we also provided the energy dispersive X-ray spectroscopy (EDS) on the nanonet or in the void hole as shown in Figure S1 in the supporting information. As expected, there are also signals of Cs, Cu and I in the EDS for the void hole. It also confirms the observed nanonet is actually bowl-like net.
According to the reference viewpoint (Adv. Funct. Mater. 2017, 27, 1603653), such inclined side wall of the bowl is facilitated to improve the light harvesting since scattering light inducing by the incident light reflected at the interface of the inclined side wall can be reabsorbed by the nanonet. In comparison, the scattering light at planar thin-film interface almost all lost without reabsorption. It is the reason why nanonet can improve the performance of photodetector. Actually, similar results were also observed in the reference work (Adv. Funct. Mater. 2017, 27, 1603653), in which CH3NH3PbI3 nanonet photodetector also had much higher responsivity and detectivity than photodetector based on CH3NH3PbI3 planar thin-film. Obviously, the better performance of photodetectors with nanonet structure than that of planar thin-film structure is independent the materials but relates to the special structure itself.
By the way, the stability experiments for nanonet photodetector were complemented in Figure S4 of the supporting information. Unlike to the high instability of lead-based perovskites, Cu-based perovskites are famous for their high stability. As a result, photodetector made by Cs3Cu2I5 nanonet also have good stability.
Revision: (1) In the second paragraph of page 8, the following discussion was added: “The improved light harvest due to scattering of incident light inside the nanonet is believed as the reason for the better performance of nanonet PD. Similar results were also reported for CH3NH3PbI3 nanonet PD and film PD.37”
(2) The energy dispersive X-ray spectroscopy (EDS) on the nanonet or in the void hole were complemented in Figure S1 in the supporting information. In the first paragraph of page 6, the following discussion has been added: “AFM result in Figure 2d provides a direct proof for the bowl morphology inside the nanonet since the side wall of the bowl is inclined to the substrate. Figure S1 shows the energy dispersive X-ray spectroscopy (EDS) at the nanonet and at the nanoholes. As expected, there are also signals of Cs, Cu and I in the EDS for the nanohole. It also confirms the observed nanonet is bowl-like net.”
(3) In the end for the first paragraph of page 9, the following discussion about the stability has been added: “Besides, the stability of Cs3Cu2I5 nanonet PD was estimated by putting PD under either thermal environment at 120 ℃ or strong ultraviolet light (275 nm, 3.5 mW/cm2) environment for 0 h, 6 h and 12 h. After that, the the nanonet photodetector was tested. From I-t curves of Figure S5 in the supporting information, the current of PD can preserve 97.8% (6 h) and 94.9% (12 h) in the 120 ℃ thermal environment or 97% (6 h) and 88% (12 h) in the ultraviolet light environment in comparison to that of freshly-fabricated PD. Obviously, Cs3Cu2I5 nanonet photodetector has high stability. ”
Comment 3: The experimental details are an important part of the manuscript so all the details should be added to the manuscript for the scientific community to follow the critical parameters. Some details are missing such as what is area of exposure, dark conditions, experimental conditions vacuum or air etc.
Reply and revision: The size of the nanonet photodetector is 1.5×1.5 cm2. During light illumination, the exposure area is believed as 1.5×1.5 cm2 since the prepared nanonet is bowl-like net as mentioned above. As-prepared PD was directly tested in the open air. The dark current was measured at a dark room without any photo illumination. During I-V measurement for photodetection, 275 nm UV light was used. In the end of Experimental section (the first paragraph of page 5), the following sentences have been added: “As-prepared PD was directly tested in the open air at room temperature. For current-voltage (I-V) measurement under photo or under dark, PD with area of 1.5×1.5 cm2 was exposed under the 275 nm light irradiation or in a dark room.”
Comment 4: The reference template is not uniform such as some references show month and year of publication, but some don’t (ref1 and others if required). One template should be followed to remove these non-uniformities. The authors are also required to cite recent papers on photodetection including Advanced Functional Materials 32 (3), 2105038 (2022) and Nanotechnology 31 (15), 152002 (2020)
Reply: We have revised all the references according to the requirements. The recommended references above have been added as Ref. 5 and 23.
